# Declines in Tapentadol Use in the US but Pronounced Regional Variation

**DOI:** 10.3390/pharmacy13030067

**Published:** 2025-05-14

**Authors:** Ching Y. Low, Kenneth L. McCall, Brian J. Piper

**Affiliations:** 1Department of Medical Education, Geisinger College of Health Sciences, Scranton, PA 18509, USA; sherynqylow@outlook.com; 2School of Medicine, Rutgers New Jersey Medical School, Newark, NJ 07103, USA; 3Department of Pharmacy Practice, Binghamton University, Johnson City, NY 13790, USA; kmccall@binghamton.edu; 4Department of Pharmacy Practice, University of New England, Biddeford, ME 04005, USA; 5Center for Pharmacy Innovation & Outcomes, Geisinger College of Health Sciences, Danville, PA 17821, USA

**Keywords:** ARCOS, diabetes, opiate, opioid, pain, pharmacoepidemiology

## Abstract

Background: Tapentadol is an atypical opioid with a dual mechanism as a mu agonist and norepinephrine reuptake inhibitor. This study characterized tapentadol use in the United States (US) using three databases. Methods: Drug distribution data from 2010 to 2020 were extracted from the Drug Enforcement Administration (DEA)’s Automated Reports and Consolidated Orders System (ARCOS), including use per region (mg/person) and business activity (i.e., pharmacy). Tapentadol prescription claims from the Medicare and Medicaid programs for 2010–2020 were also examined. Results: The distributed amount of tapentadol was 3.5 tons in 2020. Distribution was over twice as high in southern (South Atlantic = 29.0 mg/person, East South Central = 28.8) relative to Pacific (12.9) or New England (12.8) states. Tapentadol use decreased nationally between 2012 and 2020 by −53.8%. Adult diabetes prevalence was significantly associated with tapentadol distribution in 2012 (r(50) = +0.44, *p* < 0.01) and 2020 (r(50) = +0.28, *p* < 0.05). Tapentadol prescribing to Medicaid patients declined −55.2% from the peak year, 2011, until 2020. Tapentadol prescribed by Nurse Practitioners accounted for over one-sixth (18.0%) of 2019 in Medicare. Conclusions: There has been a substantial decline over the past decade in tapentadol distribution and prescribing. However, the substantial regional differences may warrant further attention by opioid stewardship programs.

## 1. Introduction

Over one million overdoses have been attributed to the US opioid epidemic [1]. There is an ongoing need for other pharmacotherapies to treat acute and chronic pain which may be less likely to be misused and diverted than selective µ opioid receptor (MOR) agonists. Tapentadol is a synthetic, centrally acting analgesic that combines agonist activity at MOR with norepinephrine reuptake inhibition (NRI). Tapentadol had similar affinity for the human norepinephrine (1.3 μM) and serotonin transporter (3.3 μM) [2]. The tapentadol immediate release (IR) formulation was approved in 2008 for acute pain in adults (≥18) severe enough to warrant an opioid analgesic, whereas tapentadol extended release (ER) was approved in 2013 for around-the-clock, long-term treatment for severe chronic pain and severe pain associated with diabetic peripheral neuropathy in the US. However, the American Diabetes Association does not recommend tapentadol as either a first or second line for diabetic neuropathy due to safety concerns such as the risk of addiction and relatively modest pain reduction [3]. Similarly, the 2020 National Institute for Health and Care Excellence guidelines on the management of neuropathic pain in adults in non-specialist settings do not address a place in therapy for tapentadol [4]. The common adverse effects of tapentadol IR were somnolence and nausea, while ER produced vomiting, constipation, and dizziness [3]. Analysis of a World Health Organization database identified 42 cases of serotonin syndrome where tapentadol was the single causative agent [2]. The US Drug Enforcement Administration (DEA) classified tapentadol as a Schedule II controlled substance [5], although tapentadol ER cost less than other Schedule II opioids when sold illicitly [6].

Prior pharmacoepidemiological research has examined typical opioids including fentanyl [7], hydrocodone [8], meperidine [9], and methadone [10], but less is known for tapentadol. The purpose of this study was to investigate the pattern in tapentadol’s distribution and prescriptions in the US from 2010 through 2020. This interval includes when there have been pronounced corrections in earlier excesses in US opioid prescribing [8] but also continued increases in obesity and diabetes. Three complementary datasets, including the Automation of Reports and Consolidated Orders Systems (ARCOS) published by the DEA [11,12], Medicaid [13], and Medicare Part D [14] programs, were used. This study also characterized the distribution of tapentadol relative to other Schedule II opioids to further understand tapentadol’s distribution in the context of additional opioid US prescribing regulations [15]. Examination of the correlation between the adult prevalence of diabetes [16] and tapentadol distribution was also determined. This report also attempted to validate ARCOS by examining the correspondence with Medicaid.

## 2. Materials and Methods

Procedures: Three complementary datasets (the DEA’s production quotas/ARCOS, Medicaid, and Medicare) were obtained for this observational report. Tapentadol and other Schedule II opioids (codeine, fentanyl, hydrocodone, hydromorphone, meperidine, morphine, oxycodone, oxymorphone) were extracted from the DEA’s ARCOS, a comprehensive drug reporting database containing an annually updated report of the distribution of DEA controlled substances from manufacturers and distributors to pharmacies and hospitals. ARCOS has been used in prior pharmacoepidemiology studies and showed a strong (r = 0.985) correlation with a state prescription drug monitoring program for oxycodone [8]. We extracted tapentadol (g) by state, quarter, and retail distribution via hospitals, pharmacies, practitioners, and teaching institutions [11]. The annual final adjusted production quota (g) of tapentadol was also extracted from the DEA [12].

Data for covered outpatient drugs paid by state Medicaid agencies have been reported to the Centers for Medicare and Medicaid Services (CMS) and captured in the State Drug Utilization Data (SDUD) database [13]. We extracted Medicaid claims for tapentadol including the corresponding 11-digit National Drug Code (NDC) for every tapentadol claim in each quarter from 2013 to 2019. We determined the milligrams per tablet from the NDC Directory [17] and thereafter computed the tapentadol (grams) reimbursed per entry. Total distributed tapentadol in grams using ARCOS and Medicaid data were converted to metric tons. Prescriber state and provider specialty type were extracted for tapentadol from Medicare Part D Prescriber Data from the Centers for Medicare and Medicaid Services from 2013 through 2019 [14], the most recent year available, to compute Medicare claims on tapentadol aggregated by prescribers’ specialty and the state the prescribers were located in.

The FDA labeling for tapentadol cautions use in pediatric patients as the safety and efficacy have not been established in this population [18]. We found no prescription orders of tapentadol to anyone younger than 18 by the Geisinger Health System, which served three million patients, between 2009 and 2020. Therefore, we calculated tapentadol (mg/person) using estimates of people aged 18 and above from 2010 to 2020 extracted from the US Census Bureau, Population Division [19].

The morphine milligrams equivalent (MME) for each Schedule II opioid was determined. Conversions were completed with the multipliers of [8] codeine, 0.15; fentanyl, 75; hydrocodone, 1; hydromorphone, 4; meperidine, 0.1; morphine, 1; oxycodone, 1.5; oxymorphone, 3; and tapentadol, 0.4, which have also been employed in prior pharmacoepidemiological reports [7,8,9,10,11]. The diabetes prevalence was retrieved to compute adult diabetes diagnosed per 100 people in each state [16]. Institutional Review Board approval was obtained from Geisinger (2021–0312, Approved 13 March 2021) and the University of New England (CR00001291, Approved 12 December 2022).

Statistics: GraphPad Prism (Boston, MA, USA, version 10.2.0: https://www.graphpad.com/ accessed on 7 May 2025) and Microsoft Excel (Redmond, WA, USA, version 2312, https://www.microsoft.com/en-us/ accessed 7 May 2025 ) were used to graph and analyze the data. We identified the peak year and expressed ARCOS and Medicaid findings as the percent change relative to this peak. Results were displayed on a heatmap (http://www.heatmapper.ca/ accessed on 7 May 2025). As has been done previously [7], percentage change values were deemed significant if ≥1.96 standard deviations above or below the mean. Tapentadol by retail distributors including hospitals, pharmacies, practitioners, and teaching institutions was expressed as the percent of the total for each year. A *t*-test between distributed tapentadol and the final adjusted production quota was performed. Regional analysis was conducted by dividing ARCOS aggregate data per US Census divisions (New England, Middle Atlantic, East North Central, West North Central, South Atlantic, East South Central, West South Central, Mountain, Pacific). The distribution pattern for tapentadol relative to that of other Schedule II opioids was determined at the national level and separately for a particularly atypical state (New Hampshire). Analysis of the correlation between ARCOS and Medicaid data was completed to provide information about the validity of ARCOS. Aggregate claims grouped by prescriber specialty and state using Medicare data were presented as percentages for each year. Adult diabetes diagnoses per 100 persons were correlated with tapentadol (mg/person aged >18) from ARCOS at the national level and within each state. A *p* < 0.05 was considered statistically significant.

## 3. Results

Between 2012, the peak year, and 2020, tapentadol saw a −53.8% reduction in the US as reported to ARCOS. In the same period, other Schedule II opioids also showed declines, with meperidine having the most reduction (−91.8%), and codeine the least (−32.1%; Figure 1A). On average, the final adjusted quota from ARCOS was 207.2% greater than the distributed amount, and this difference was significant (Figure 1B).

The distribution of Schedule II opioids in MME determined that oxycodone accounted for nearly half of 2020’s distribution, while tapentadol accounted for 1.3%. The percent distribution of opioids was relatively constant in 2012 and 2020 (Figure 1C). The amount of tapentadol distributed by each business activity showed modest variation between 2010 and 2020. Pharmacies were consistently the predominant distribution channel (98.0% in 2010 and 98.1% in 2020). Hospitals increased from 1.1% to 1.9% of distribution, and this gain was achieved with a corresponding decrease among practitioners (Figure 1D).

A regional analysis revealed that tapentadol distribution to New England and the Pacific states was less than half of the East South Central and South Atlantic states (Figure 2A). Exploratory analyses determined that the New England and Pacific states were significantly lower than all other US Census divisions except the West North Central (Figure 2B).

Every state in the US saw a reduction in tapentadol between 2012 and 2020, except for New Hampshire (NH, +13.1% increase, Figure 3A). The percent changes for Idaho (−14.7%) and NH were outside the 95% confidence interval (Figure 3B). A correlation analysis on ARCOS with Medicaid showed a strong, positive, and significant correlation of 0.80 (Figure 4A).

Tapentadol prescription claims from Medicaid mirrored the distribution reported by ARCOS. Medicaid prescriptions peaked in the second quarter of 2011, and there was an appreciable decrease (−55.2%) from then until the end of 2020 (Figure 4B).

Next, tapentadol claims from Medicare Part D were presented as a percentage for each year by prescribing specialties. Nurse Practitioners experienced a strong and steady increase in their share of tapentadol’s prescriptions and became the specialty with the largest portion in 2019. NPs accounted for 9.3% in 2013 but doubled to 18.0% in 2019. Physician Assistants experienced a steady but more moderate increase from 9.3% in 2013 to 14.5% in 2019 (Figure 5).

Because NH was atypical in ARCOS, exploratory analyses were completed for this state. Certified Registered Nurse Anesthetists (CRNAs) saw a steep increase from 16.9% in 2013 to 59.7% in 2015, followed by a pronounced decline to 5.3% in 2019. Nurse Practitioners showed a strong increase in the share of prescriptions after the dip between 2013 and 2016, where it became the specialty with the largest share at 36.0% in 2019. Pain management experienced a seven-fold increase from 1.9% in 2013 to 13.7% in 2019 (Figure 6).

Tapentadol’s distribution in ARCOS and the diabetes prevalence showed a moderate correlation of 0.44 (*p* < 0.01) for 2012. Similarly, it was 0.28 (*p* < 0.05) for 2020 (Figure 7).

## 4. Discussion

This novel study identified pronounced, but regionally dependent, changes in the prescribing pattern of tapentadol in the US between 2010 and 2020, with a notable peak in 2012. Using ARCOS data, tapentadol decreased −53.8% between 2012 and 2020, and its decrease was ranked fourth among Schedule II opioids (Figure 1A). Given that tapentadol is mechanistically unusual for a Schedule II opioid [2] and is approved to treat neuropathic pain associated with adult diabetes, a condition which has increased over the past decade, we wanted to better understand these use patterns. Distribution of tapentadol to New England states was less than half that in the southern US. Significant associations between diabetes prevalence and tapentadol distribution were also observed (Figure 7). This correlation was slightly lower in 2020 (r = 0.28) than 2012 (r = 0.44), which might reflect increased use of non-Schedule II alternatives like pregabalin or duloxetine [3] for neuropathic pain among diabetes patients. Similarly, a prior study found that meperidine distribution was moderately (r = 0.48) associated with state level of obesity, with the highest distribution in Arkansas, Alabama, Mississippi, and Louisiana [9].

There were notable changes in the insurance and policy intervention landscapes in the period surrounding 2012, likely in response to a November 2011 communication from the Centers for Disease Control and Prevention stating that overdoses from prescription opioid painkillers had reached epidemic levels [20]. States adopted policy interventions such as mandated Prescription Drug Monitoring Programs (PDMPs) and pain clinic laws, while the Centers for Medicare and Medicaid Services implemented the Overutilization Monitoring System in 2013 [21]. In addition, private insurers began to take actions to limit the reimbursement of opioids [22]. We recognize success in containing the opioid crisis varies, as do state-level policy interventions, insurance reimbursements, prescription rights, and prescribing behaviors [7,23,24,25].

Our data showed that all states in the US showed a reduction in the percentage change in the amount of tapentadol distributed between 2012 and 2020, except NH (+13.1%), and this was statistically significantly different from the national average (Figure 2B). New Hampshire providers wrote 46.1 opioid prescriptions for every 100 persons in 2018, relative to the average US rate of 51.4 prescriptions [26]. However, tapentadol and methadone were two opioids that did not show similar declining patterns as other opioids. Methadone is approved by the US Food and Drug Administration (FDA) for treating opioid use disorder (OUD), although some formulations are also used for pain [10]. New Hampshire has been expanding the availability of addiction treatment through medication-assisted treatment for OUD [27]. A purported off-label use of tapentadol may include it being used as part of a step-down therapy in treating OUDs [28]. More than 97% of tapentadol distributed in the US from 2010 through 2020 were through pharmacies. This indicates that most tapentadol was prescribed in outpatient services and office settings (Figure 1D). Tapentadol distributed by hospitals steadily increased to nearly 2% of the total in 2020, and this observation can be partly explained by anesthesiology as a specialty increasing in its share (13.8%) of the total tapentadol Medicare claims in 2019 (Figure 5). A randomized controlled trial in Australia determined that restricting oxycodone in the emergency room resulted in a doubling of tapentadol [29]. Surgery, neurology, and spinal rehabilitation were the most common specialties for tapentadol prescribing in the Prince of Wales hospital in Sydney [30].

For both US and NH, Nurse Practitioners (NPs) were the specialty that had the largest share of tapentadol prescribed using Medicare Part D data in 2019. NPs prescribed 18.0% of Medicare claims of tapentadol at the national level (Figure 5) and 36.0% in NH (Figure 6). NPs increased the number of opioids they prescribed following a grant of NP independence [31], and NPs and Physician Assistants (PAs) practicing in states with independent prescription authority had a greater number of outliers who prescribed high-frequency, high dose opioids than did MDs [32]. Prescriptive rights of NPs differ across states. In NH, NPs were granted full authority to prescribe Schedule II to IV controlled substances in 2009 [33]. Our data showed that NPs and CRNAs experienced opposing trends between 2013 and 2019. NPs decreased sharply from 57.1% in 2013 to 10.8% in 2016 then rebounded strongly to 36.0% and ranked first among all specialties in 2019. According to the American Association of Nurse Practitioners, 70.2% of all NPs deliver primary care in 2020 [34], and therefore there is an increased likelihood of NPs caring for chronic but less complex cases.

It was also noteworthy that tapentadol production quotas [12] were double (207.2%) distribution. The DEA’s final adjusted quota seeks to “ensure an adequate and uninterrupted supply of controlled substances in order to meet the demand of legitimate medical, scientific, and export needs of the United States” [12]. Our data suggest the DEA’s quota was not a determining factor influencing the amount of tapentadol distributed in the US. Presumably, this was because the DEA’s production quota also included tapentadol which was subsequently exported to other countries.

Tapentadol distribution, when expressed as MME, accounted for a modest portion (1.3%) of the 2020 total for Schedule II substances. However, this amount was greater than that of oxymorphone or meperidine [9]. Meperidine [35], like tapentadol, has been linked to cases of serotonin syndrome [36]. Also similar to meperidine, there have been concerns with tapentadol and neurotoxicity [37]. The atypical mechanism of tapentadol may offer another advantage relative to typical opioids in not inhibiting neurogenesis, at least in rodents [38], but this awaits other human research for verification. The amount of tapentadol reported to the DEA’s ARCOS strongly and significantly correlated with the amount of Medicaid claims, at 0.80 (Figure 4A), suggesting that tapentadol was employed at a population level broadly similarly for Medicaid and non-Medicaid patient populations.

Some caveats and future directions should also be considered. A limitation of ARCOS is that tapentadol was reported by weight, with no information on the number of prescriptions or recipients. However, a strength of this report relative to previous ones [7,8,9,10] is the inclusion of prescription information from the Medicaid and Medicare programs. These databases are complementary in providing population-level patterns, but they are not generalizable to patient-level patterns. The high correspondence between ARCOS and Medicaid indicates that opioid distribution is a valid population level measure. Not all analyses (e.g., the comparison of New England vs. other regions) were hypothesized a priori and should be interpreted with caution. Given that tapentadol is not recommended as a first, or even second, line pharmacotherapy for neuropathic pain [3], follow-up studies with electronic health records examining multiple pharmacotherapies may be necessary as the rates of adult diabetes continues to increase [39]. Electronic health records will also be valuable to further characterize whether prescriptions expand to other pain conditions (e.g., fibromyalgia) [40] or to children and adolescents [41]. The limited misuse of tapentadol relative to other Schedule II opioids [6] may not be a sufficient advantage relative to another atypical opioid, tramadol, which is Schedule IV.

Recently, the German drugmaker Grünenthal was accused of conducting a disinformation campaign by promoting this Schedule II opioid [42] as “not an opioid”, claiming that tapentadol has “minimum potential of abuse”, causes less dependence than other opioids, and “has no street value” and minimizing the potential for respiratory depression [43]. Several converging lines of evidence have identified dopamine as integral to addiction, including for cocaine, amphetamine, nicotine, ethanol, and morphine [44]. Intraperitoneal administration of tapentadol to rats evoked a pronounced (>600%) release of cortical dopamine [45]. The street value of tapentadol from StreetRx.com in 2016 was USD 0.18/mg [6] which, for the maximum daily dose (600 mg), would result in a non-trivial yearly diversion value (USD 39,420). A systematic review of studies published though 2017 identified ten overdoses involving tapentadol polysubsance use and four with only tapentadol [36]. This included a 34-year-old Montana man with a history of IV drug use found gasping for breath by his wife. He did not have a prescription for tapentadol, and analytical chemistry did not detect any other substances of significance to this overdose [46]. A 40-year-old man with a history of depression, anxiety, and a shoulder injury and a recent tapentadol prescription had heart blood tapentadol levels at over twenty-fold higher than the upper limit of the therapeutic range [47]. These issues will warrant continued attention by opioid stewardship committees and others responsible for the judicious use of prescription opioids.

## 5. Conclusions

This study quantified the dynamic pattern in tapentadol distribution from 2010 to 2020. Distributed tapentadol peaked in 2012. Thereafter, there has been a reduction in the distribution of tapentadol at both national and state levels. The declining tapentadol pattern is consistent with other opioids because, since 2011 [7], there has been a steady decrease in the distribution of most medications used for pain [8,9,10]. Tapentadol use showed clear regional differences in distribution which were significantly associated with diabetes prevalence. As tapentadol is the only Schedule II opioid that has dual MOR and monoamine transporter mechanisms, the role of this atypical opioid should continue to be evaluated, particularly within the framework of a disinformation campaign from the manufacturer downplaying the risk of addiction [43], as well as the evolving US obesity and opioid epidemics.

## Figures and Tables

**Figure 1 pharmacy-13-00067-f001:**
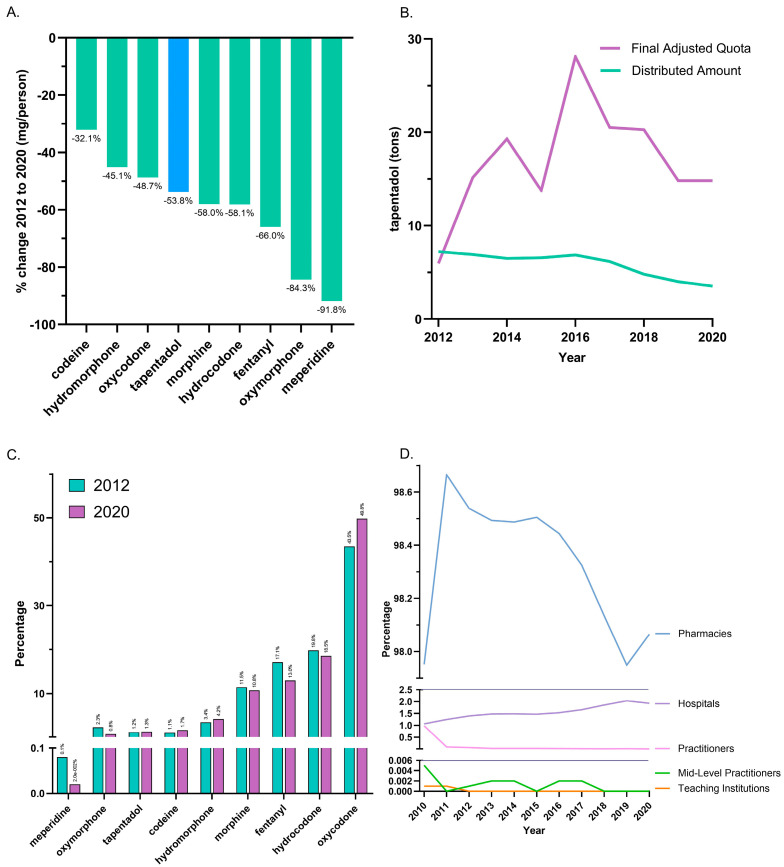
Controlled substance distribution for tapentadol and other opioids from the Drug Enforcement Administration’s Automated Reports and Consolidated Ordering System. (**A**) All Schedule II opioids declined, and tapentadol decreased by 53.8% from 2012 until 2020. (**B**) Comparison of DEA production quota and distribution. A *t*-test between distributed and final adjusted quota amounts of tapentadol was statistically significant (*p* < 0.05) for 2012–2020. (**C**) Percentages of distributed Schedule II opioids in morphine mg equivalents (MME) for 2012 and 2020. (**D**) Tapentadol by retail distributors as a percentage of the total for 2010–2020.

**Figure 2 pharmacy-13-00067-f002:**
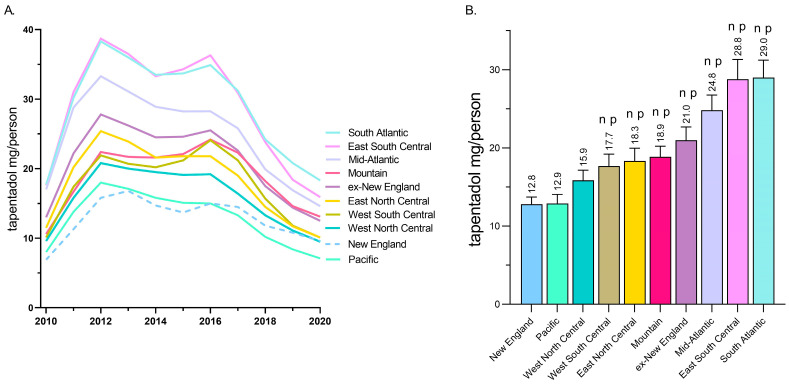
Tapentadol (mg/person aged 18 and above) in various regional divisions using data from the Drug Enforcement Administration’s Automated Reports and Consolidated Ordering System. (**A**) Time-series. (**B**) Regional analysis. Regions indicated with an n were statistically different from New England, whereas regions indicated with p were statistically different from Pacific (*p* < 0.05).

**Figure 3 pharmacy-13-00067-f003:**
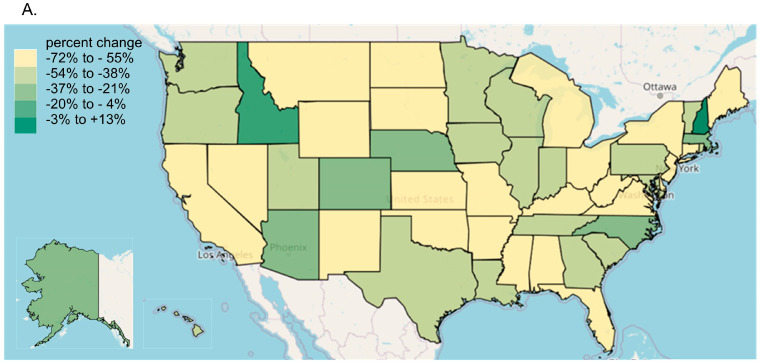
Tapentadol distribution from the Drug Enforcement Administration’s Automated Reports and Consolidated Ordering System. (**A**) Heatmap of percent change (mg/person) between 2012 and 2020. Only NH experienced a positive change (+13.1%). (**B**) Percent change between 2012 and 2020. The mean was −51.7%, designated with a vertical dashed line, and the SD was 16.1%. States with a percentage change that was significantly different than the national average (* *p* < 0.05).

**Figure 4 pharmacy-13-00067-f004:**
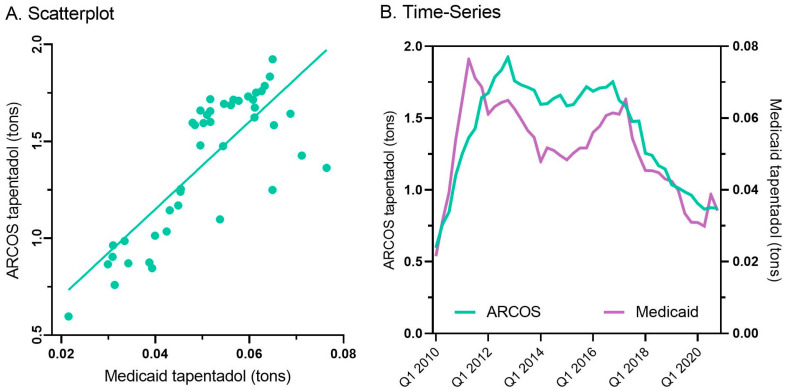
Scatterplot (r(50) = +0.80, *p* < 0.001) (**A**) and quarterly (**B**) prescribed tapentadol from Medicaid and distribution from ARCOS.

**Figure 5 pharmacy-13-00067-f005:**
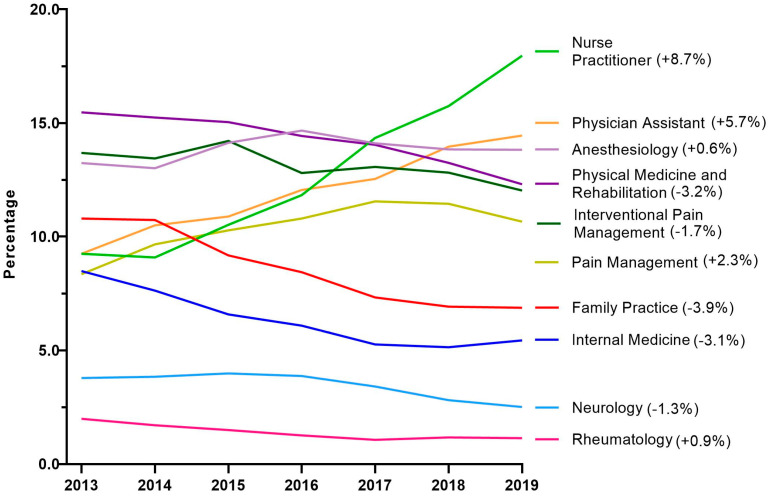
Percent of Medicare Part D tapentadol claims by prescriber specialty. The difference from 2013 to 2019 is in parentheses after each specialty.

**Figure 6 pharmacy-13-00067-f006:**
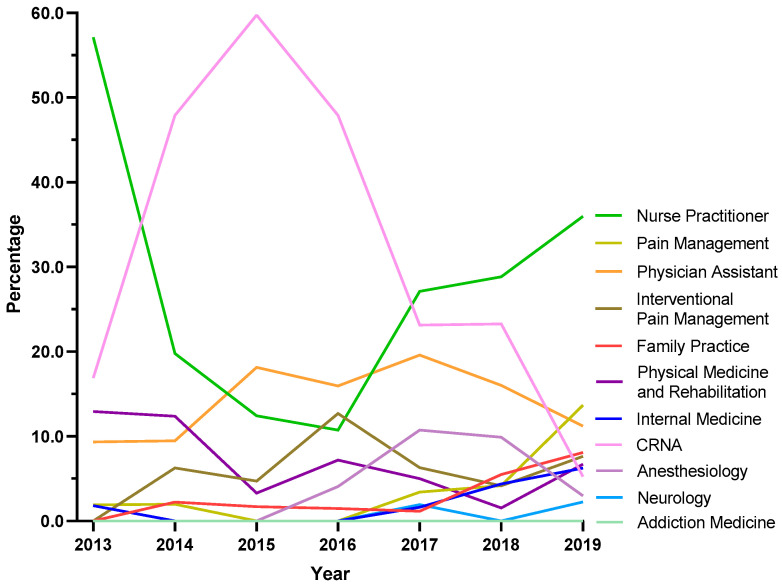
Tapentadol claims in New Hampshire from Medicare Part D expressed as a percentage of each year by prescriber specialty. Nurse Practitioners prescribed 36.0% Medicare claims of tapentadol in 2019, followed by Pain Management (13.7%) and Physician Assistants (11.2%).

**Figure 7 pharmacy-13-00067-f007:**
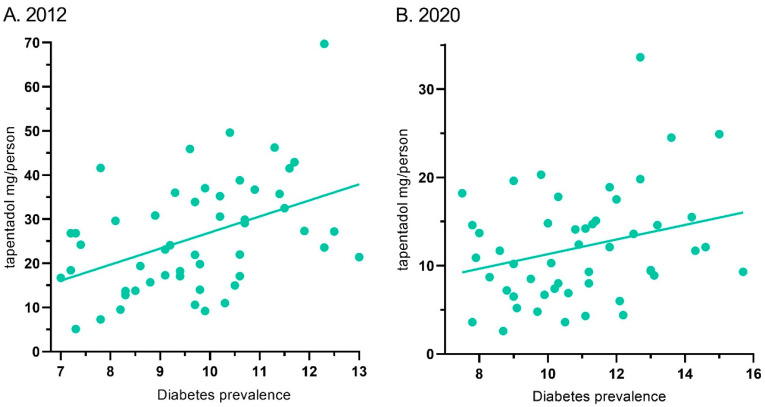
Correlation between diabetes prevalence and tapentadol distribution for 2012 ((**A**) r(50) = 0.44, *p* < 0.01) and 2020 ((**B**) r(50) = 0.28, *p* < 0.05).

## Data Availability

All data is publicly available from the ARCOS, Medicaid, and Medicare databases. ARCOS data is available at https://www.medrxiv.org/content/10.1101/2022.03.03.22271869v1.supplementary-material (accessed on 7 May 2025). Medicaid is available at https://www.medicaid.gov/medicaid/prescription-drugs/state-drug-utilization-data/index.html (accessed on 7 May 2025). Medicare Part D is available at https://data.cms.gov/provider-summary-by-type-of-service/medicare-part-d-prescribers/medicare-part-d-prescribers-by-geography-and-drug (accessed on 7 May 2025).

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
