# Peer review of "Declines in Tapentadol Use in the US but Pronounced Regional Variation"

_pharmacy, 2025, doi:10.3390/pharmacy13030067_

Round 1
Reviewer 1 Report
Comments and Suggestions for Authors
Thank you kindly for the opportunity to review your manuscript on tapentadol prescribing.
Overall, I find this to be an easy read with clear results.
Thank you also for numbering the lines in your manuscript, that makes it so much easier for us to communicate.
Please review Lines 60 and 69. In line 60 you tell us there are 3 datasets and in line 69 there are 4. Also check complementary and complimentary, I suspect that none of the data sets offered compliments and that complementary is the word you want.
on line 95 please verify verb tense, I believe you intended to say served
on line 105 are the words "approved date" merely place holders and you intended to include a date?
I truly appreciate figure 3, it certainly helps to illustrate your discussion
I do not find figure 6 helpful, but some readers might
Starting line 242 I am interested in where emergency room visits fall in your description. Are they "outpatient services"? Many rural emergency rooms do NOT dispense take home meds so the results would be a prescription and I don't feel as if that has been captured here.
the supplemental Figure 1 on line 332 is important to illustrating your comments regarding New Hampshire and I would like to see it more prominent in your manuscript rather than tucked in at the end.
Author Response
We would like to thank Reviewer #1 for the detailed and timely feedback. Adjustments to the manuscript are highlighted in green. Please note that responses to Reviewer #2 are highlighted in blue.
Thank you kindly for the opportunity to review your manuscript on tapentadol prescribing.
Overall, I find this to be an easy read with clear results.
Thank you also for numbering the lines in your manuscript, that makes it so much easier for us to communicate.
- Please review Lines 60 and 69. In line 60 you tell us there are 3 datasets and in line 69 there are 4.
Good catch! Yes, there are different components of the DEA database. Line 69 was changed from four to three to be more consistent.
- Also check complementary and complimentary, I suspect that none of the data sets offered compliments and that complementary is the word you want.
Yes, changed line 60 to complementary.
- on line 95 please verify verb tense, I believe you intended to say served
Yes, changed verb tense as suggested from serve to served (current line 96).
- on line 105 are the words "approved date" merely place holders and you intended to include a date?
Correct. The date of approval has now been included: 12/6/2022.
- I truly appreciate figure 3, it certainly helps to illustrate your discussion. I do not find figure 6 helpful, but some readers might.
Yes, Figure 6 fits with our broader program of research that attempts to examine use of various pharmacotherapies and whether there is any correspondence, at a population level, with any diseases or conditions.
- Starting line 242 I am interested in where emergency room visits fall in your description. Are they "outpatient services"? Many rural emergency rooms do NOT dispense take home meds so the results would be a prescription and I don't feel as if that has been captured here.
Yes, this is a key point. Unfortunately, our datasets do not perfectly address this. Although we reported on Medicare Part D claims, opioids from a stay in a hospital (or skilled nursing facility) could be reported by Medicare Part B. Yes, tapentadol prescribed by the ER provider but filled in a retail pharmacy would be reported by ARCOS under pharmacy. We have now included some discussion of two new studies that better address this issue: “A randomized-controlled trial in Australia determined that restricting oxycodone in the emergency room resulted in a doubling of tapentadol [29]. Surgery, neurology, and spinal rehabilitation were the most common specialties for tapentadol prescribing in the Prince of Wales hospital in Sydney [30].
New citations
- Mitra B, Roman C, Wu B, Luckhoff C, Goubrial D, Amos T, Bannon-Murphy H, Huyhn R, Dooley M, Smit DV, et al. Restriction of oxycodone in the emergency department (ROXY-ED): A randomized controlled trial. Brit J Pain 2023; 17(5):491-500.
- Mirabella J, Ravi D, Chiew AL, Buckley NA, Chan BS. Prescribing trend of tapentadol in a Sydney health district. Brit J Clin Pharmacol. 2022; 88:3929-3935.
5. the supplemental Figure 1 on line 332 is important to illustrating your comments regarding New Hampshire and I would like to see it more prominent in your manuscript rather than tucked in at the end.
Good point. Sup Fig 1 has been relocated to Figure 6.

Reviewer 2 Report
Comments and Suggestions for Authors
After critically reviewing this Research Article titled "Declines, but pronounced regional variation, in tapentadol use in the US", I detected some MINOR flaws, which determined my recommendation of “ACCEPT UNDER MINOR REVIEW”. Below please find my detailed comments.
The authors set out to conduct a study that characterized the use of tapentadol in the United States (US) using three databases: drug distribution between 2010 and 2020 was extracted from the Drug Enforcement Administration’s (DEA) Automated Reports and Consolidated Orders System (ARCOS). Considering the importance of the irrational use of opioid drugs and the high percentage of deaths due to their use, the study proves to be quite relevant.
The introduction is well written, but could add other indications of tapentadol based on the literature. The results are many, but the way they were described, in this reviewer's opinion, is clear.
My suggestions are below:
- In the abstract, the authors state that they consulted three databases "Drug distribution from 2010-2020 were extracted from the Drug Enforcement Administration’s (DEA) Automated Reports and Consolidated Orders System (ARCOS)", but at the beginning of the Material and Methods chapter it is described: "Procedures: Four complementary datasets (DEA’s production quotas, DEA’s ARCOS, Medicaid, and Medicare) were obtained for this observational report". The authors should correct which information is correct regarding this.
- In the introduction it would be important to at least mention the other uses of tapentadol, in addition to diabetic neuropathy.
- In the discussion, last sentence: “There is also an increasing, but small, literature of non-oral tapentadol misuse and overdoses [35-37]”. I would like the subject to be a little more detailed.
- There are several other abbreviations in the text that were not described at the end of the paper, the authors should update the list.
The paper's conclusions are in line with the findings and well written.
Author Response
Dear Reviewer #2,
Thank you for the detailed and insightful feedback. Responses are below to the points made (now numbered) and modifications to the manuscript are highlighted in blue. Modifications made in response to Reviewer #1 are in green.
Brian
- After critically reviewing this Research Article titled "Declines, but pronounced regional variation, in tapentadol use in the US", I detected some MINOR flaws, which determined my recommendation of “ACCEPT UNDER MINOR REVIEW”. Below please find my detailed comments.
The authors set out to conduct a study that characterized the use of tapentadol in the United States (US) using three databases: drug distribution between 2010 and 2020 was extracted from the Drug Enforcement Administration’s (DEA) Automated Reports and Consolidated Orders System (ARCOS). Considering the importance of the irrational use of opioid drugs and the high percentage of deaths due to their use, the study proves to be quite relevant.
The introduction is well written, but could add other indications of tapentadol based on the literature.
We have included other indications based on the literature. For stylistic/flow reasons, this was added to lines 301-3: “Electronic health records will also be valuable to further characterize whether prescriptions expand to other pain conditions (e.g. fibromyalgia) [#] or to children and adolescents [#].”
New citations:
Littlejohn GO, Guyner EK, Nglan GS. Is there a role for opioids in the treatment of fibromyalgia? Pain Manag 2016; 6:347-355.
Eerdekens M, Radic T, Sohns M, Khalil F, Bulawa B, Elling C. Outcomes of the pediatric development plan of tapentadol. J Pain Res 2021; 14:249-61.
- The results are many, but the way they were described, in this reviewer's opinion, is clear.
My suggestions are below:
In the abstract, the authors state that they consulted three databases "Drug distribution from 2010-2020 were extracted from the Drug Enforcement Administration’s (DEA) Automated Reports and Consolidated Orders System (ARCOS)", but at the beginning of the Material and Methods chapter it is described: "Procedures: Four complementary datasets (DEA’s production quotas, DEA’s ARCOS, Medicaid, and Medicare) were obtained for this observational report". The authors should correct which information is correct regarding this.
Good catch (also noted by Reviewer #1). This has been changed as suggested on Line 70 to “three”.
- 3. In the introduction it would be important to at least mention the other uses of tapentadol, in addition to diabetic neuropathy.
The intro has been expanded (line 40) to include: “The tapentadol immediate release (IR) formulation was approved in 2008 for acute pain severe in adults (> 18) …”
- 4. In the discussion, last sentence: “There is also an increasing, but small, literature of non-oral tapentadol misuse and overdoses [35-37]”. I would like the subject to be a little more detailed.
A new paragraph has been added to more fully contextualize information about tapentadol misuse and overdoses as:
The German drugmaker Grünenthal has been accused of conducting a disinformation campaign by promoting this Schedule II opioid as “not an opioid”, minimizing the potential for respiratory depression, and making claims that tapentadol has “minimum potential of abuse”, causes less dependence than other opioids, and “has no street value” [#]. Several converging lines of evidence have identified dopamine as integral to addiction including for cocaine, amphetamine, nicotine, ethanol, and morphine [#]. Intraperitoneal administration of tapentadol to rats evoked a pronounced ( > 600%) release of cortical dopamine [#]. The street value of tapentadol from StreetRx.com in 2016 was $0.18/mg [#] which, for the maximum daily dose (600 mg) would result in a non-trivial yearly diversion value ($39,420). A systematic review of studies published though 2017 identified ten overdoses involving tapentadol polysubsance use and four with only tapentadol [#]. This included a 34 year-old Montana man with a history of iv drug use found gasping for breath by his wife. He did not have a prescription for tapentadol and analytical chemistry did not detect any other substances of significance to this overdose [#]. A 40-year-old man with a history of depression, anxiety, and a shoulder injury had heart blood tapentadol levels at over twenty fold higher than the upper limit of the therapeutic range [#]. These issues will warrant continued attention by opioid stewardship committees and others responsible for the judicious use of prescription opioids.
New citations:
#. Solinas M, Belujon P, Fernagut PO, Jaber M, Thiriet N. Dopamine And Addiction: What Have We Learned From 40 Years Of Research. J Neural Transm 2018; 126:481-516.
#. Benade V, Nirogi R, Bhyrapuneni G, Daripelli S, Ayyanki G, Irappanavar S, Ponnamaneni R, Manoharan A. Mechanistic Evaluation Of Tapentadol In Reducing Pain Perception Using The In-vivo Brain And Spinal Cord Microdialysis In Rats. Eur J Pharmacol 2017; 809:224-30.
#. Jutkiewicz EM, Traynor JR. Opioid Analgesics. In: Brunton LL, Knollmann BC. eds. Goodman & Gilman's: The Pharmacological Basis of Therapeutics, 14th Edition. McGraw-Hill Education; 2023. Accessed May 04, 2025. https://accessmedicine-mhmedical-com.gcsom.idm.oclc.org/content.aspx?bookid=3191§ionid=269719481.
- There are several other abbreviations in the text that were not described at the end of the paper, the authors should update the list.
Several additional abbreviations were added as suggested:
CRNA: Certified Registered Nurse Anesthetist
DEA: Drug Enforcement Administration
ER: Extended Release
FDA: Food & Drug Administration
MME: Morphine Mg Equivalent
NDC: National Drug Code
